# Therapeutic Efficacy and Safety of Intense Pulsed Light for Refractive Multiple Recurrent Chalazia

**DOI:** 10.3390/jcm11185338

**Published:** 2022-09-11

**Authors:** Reiko Arita, Shima Fukuoka

**Affiliations:** 1Itoh Clinic, Saitama 3370042, Japan; 2Lid and Meibomian Gland Working Group, Saitama 3370042, Japan; 3Omiya Hamada Eye Clinic, Omiya, Saitama 3300854, Japan

**Keywords:** chalazion, intense pulsed light, meibomian gland, meibomian gland dysfunction, blepharitis

## Abstract

To evaluate the efficacy and safety of intense pulsed light (IPL) combined with meibomian gland expression (MGX) for the treatment of refractory multiple and recurrent chalazia without surgery or curettage. This was a retrospective controlled study. Patients with multiple and recurrent chalazia, who had performed the conventional treatment at least 2 months without any surgery or curettage, were enrolled in this study. Twenty-nine consecutive multiple recurrent chalazia (12 patients) were assigned to receive either the combination of IPL and MGX or MGX alone as a control. Each eye underwent one to four treatment sessions with 2-week intervals. Parameters were evaluated before and 1 month after the final treatment session. Clinical assessments included symptom, size of each chalazion, lid margin abnormalities, corneal and conjunctival fluorescein staining, meibum grade, the number of Demodex mites, the Schirmer value and meiboscore. All parameters except meiboscore and the Schirmer value were significantly improved with IPL-MGX therapy, whereas only meibum grade was significantly improved with MGX alone. There were no adverse events which occurred in either group. IPL-MGX was safe and effective for multiple and recurrent chalazia without surgery or curettage by reducing the size of chalazion and improving lid margin abnormalities and meibum grade.

## 1. Introduction

Chalazion is a non-infectious chronic granulomatous inflammation of the lipids of the meibomian glands [1,2]. Blepharitis [3] and ocular demodicosis [4] are risk factors for chalazia. Patients with demodicosis tend to demonstrate recurrence [4,5]. MGD, blepharitis and marginal keratitis are significantly associated with a higher rate of developing multiple chalazia in pediatric patients [6]. Traditionally, the “cutting” treatment of chalazion by incision and curettage has been the mainstay of treatment [7], but “non-surgical” treatment of chalazion is attracting attention in order to protect the morphology and function of the meibomian glands [1,2]. Non-surgical common therapies for patients with chalazion include various topical medications such as steroidal injection [8,9], oral and/or topical antibiotics (eyedrops and/or ointment) [10], and topical steroids (eyedrops and/or ointment) [10] as well as warming eyelids and/or lid hygiene [1,10]. While some cases are cured spontaneously with the non-surgical conventional therapies above, some cases are refractory with multiple or recurrent episodes. Chalazion is considered a pathognomonic localized form of meibomian gland dysfunction (MGD) [11]. Chalazion is a risk factor of meibomian gland loss [12]. Indeed, many cases with multiple or recurrent episodes present with MGD, which show lid margin abnormalities such as plugging and vascularity of orifices and large loss of meibomian glands. Adding surgical resection to multiple recurrent chalazion tumors not only carries the risk of recurrence, but also the concern of losing even more of the meibomian gland area, resulting in dry eye in the future [13].

Intense pulsed light (IPL) therapy based on the delivery of intense pulses of noncoherent light with wavelengths of 500 to 1200 nm has been applied in dermatology to treat various conditions, including benign cavernous hemangiomas or venous malformations, telangiectasia, port wine stains, and other pigmented lesions [14]. IPL has been published internationally for the treatment of MGD and is listed as a Step 2 treatment in the International Guidelines for Dry Eye [15]. More than seventy papers and 14 randomized controlled trials have been conducted on MGD to date [16] and the treatment has been proven to be useful. IPL is a treatment that can be a fundamental treatment for MGD. A systematic review found that IPL is an effective and well-tolerated treatment option for a range of dermatologic conditions, having been shown to result in a reduction in the extent of telangiectasia and the severity of facial erythema [17]. Side effects of IPL ophthalmic treatment include redness, swelling, hair removal [18], and although rare, pupillary constriction, anterior uveitis, and pupillary defect have been reported [19]. The efficacy of IPL therapy for patients with dry eye due to MGD was discovered during IPL treatment of facial rosacea [20]. Subsequent studies found that IPL, with or without concomitant meibomian gland expression (MGX), is effective for improvement of subjective symptoms and objective findings in patients with mild to moderate MGD or dry eye [21,22,23,24,25,26,27,28,29,30,31,32]. The combination of IPL and MGX was also shown to be effective in patients with refractory MGD [32,33].

Recently, the efficacy of IPL has been reported for recurrent chalazion after excisional surgery in a single-arm study [34]. Still, there are no reports on the treatment of multiple and recurrent chalazia using IPL without surgery, with a control group.

Although the usefulness of IPL for chalazion has been investigated, there is still no international consensus on specific indications and protocols. Based on this study, we would like to propose a protocol for IPL for multiple and recurrent chalazia. We have, therefore, performed a retrospective controlled study to evaluate the efficacy of IPL combined with MGX in patients with refractory multiple recurrent chalazia who have been treated with non-surgical conventional therapies for at least 2 months. In addition, we analyzed the factors associated with the number of IPL treatment sessions until improvement.

## 2. Materials and Methods

### 2.1. Patients

This retrospective cohort study was approved by the Institutional Review Boards of Itoh Clinic (IRIN-202109), and it adhered to the tenets of the Declaration of Helsinki. Patients with refractory multiple recurrent chalazia who were treated with IPL and MGX or MGX alone between April and December 2021 at Itoh Clinic in Japan were assigned in the study. Informed consent was obtained from all subjects involved in the study. Inclusion criteria included: (1) an age of at least 18 years; (2) a diagnosis of multiple and recurrent chalazia which occurred within 6 months in more than 2 chalazions; (3) refractoriness of chalazion as defined by the failure to respond over a period of ≥2 months to at least three types of conventional therapy prescribed in Japan, including warming eyelids, lid hygiene, topical antibiotics eyedrops and/or ointment, topical steroid eyedrops and/or ointment, and/or systemic antibiotics oral medication; and (4) a Fitzpatrick skin type of 1 to 4 based on sun sensitivity and appearance [35]. Exclusion criteria included the presence of active skin lesions, skin cancer, or other specific skin pathology or of active ocular infection or ocular inflammatory disease. 

### 2.2. Experimental Design

Patients treated with IPL-MGX underwent a series of one to four IPL-MGX treatment sessions at 2-week intervals and patients with MGX alone received treatment four times at 2-week intervals. Both groups were subjected to clinical assessment as described below both before treatment as well as 4 weeks after the final treatment session. Six months later, recurrence and safety were confirmed (Figure 1). Since this was an exploratory study, the number of IPL sessions was not determined, and the IPL was terminated when the size of the chalazion was reduced by 80–100% and the Standard Patient Evaluation of Eye Dryness (SPEED) score was less than 6 [36]. All of the patients were asked to continue warming eyelids and lid hygiene as well as not to initiate therapy with a new topical or systemic agent during the treatment course.

### 2.3. Clinical Assessment

The diameter (mm) of each chalazion was measured with a ruler and we took a photograph with a slit lamp. Lid margin abnormalities (plugging of meibomian gland orifices and vascularity of lid margins) [37], corneal and conjunctival fluorescein staining (CFS) [38], and meibum grade [39] were evaluated with a slit lamp microscope. Morphological changes of meibomian glands were assessed on the basis of the meiboscore [40] as determined by noninvasive meibography. Tear fluid production was measured by Schirmer’s test without anesthesia [41]. The number of Demodex mite was counted using a light microscope after pulling out three lashes. Symptoms were assessed with the SPEED validated questionnaire (scale of 0 to 28) for both eyes [36,42] and with visual analogue scale (VAS) scores of 0 (no symptom) to 100 (maximum conceivable symptom) of ocular discomfort and foreign body sensation for each eye separately. Visual acuity, intraocular pressure, lens opacity as well as fundus examination were also examined before and at 1 month after the final treatment session. We checked for recurrence of chalazion up to 6 months.

### 2.4. IPL-MGX Procedure

Before the first treatment, each patient underwent Fitzpatrick skin typing [35] and the IPL machine (AQUA CEL; Jeysis, Seoul, South Korea) was adjusted to the appropriate setting (upper eyelid; 15 J/cm^2^, lower eyelid; 20 J/cm^2^). At each treatment session, both eyes of the patient were closed and sealed with disposable eye shields (AQUA CEL HYDROGEL EYE CARE PATCH, KBM Inc., Seoul, Korea). After generous application of ultrasonic gel to the targeted skin area, each patient received ~13 pulses of light (with slightly overlapping applications) from the right preauricular area, across the cheeks and nose, to the left preauricular area, reaching up to the inferior boundary of the eye shields. Then, IPL was applied to the upper orbit along the bottom of the eyebrow from the temple to the base of the nose. These procedures were then repeated in a second pass. Immediately after the IPL treatment, MGX was performed on both upper and lower eyelids of each eye with an Arita Meibomian Gland Expressor (M-2073, Inami, Tokyo, Japan). Pain was minimized during MGX by the application of 0.4% oxybuprocaine hydrochloride to each eye.

### 2.5. Statistical Analysis

Data were found to be nonnormally distributed with the Shapiro–Wilk test (*p* < 0.05), and nonparametric testing was therefore applied. Wilcoxon rank sum test was used to compare the background characteristics of patients between the IPL-MGX and the MGX alone groups. Numerical data were compared between before and after treatment with the use of the Wilcoxon signed-rank test. Wilcoxon rank sum test was applied to compare the parameters between the IPL-MGX and the MGX alone groups. The number of IPLs required to improve the chalazion was investigated with background characteristics of the patients and each parameter using the Spearman’s rank correlation coefficient. 

We performed a statistical power analysis for both size of chalazion and the SPEED score. For the size of chalazion, the mean difference between before and four weeks after the final treatment was 4.3, with a corresponding standard deviation (SD) of 5.0; for the SPEED score, the mean difference was 5.4 with an SD of 6.3. For the size of chalazion, the mean difference between the IPL-MGX and the MGX groups after treatment was 9.2, with a corresponding SD of 1.9; for the SPEED score, the mean difference was 12.1 with an SD of 1.4. These changes were calculated from the results of all 12 eyes in the current study. The number of eyes in each group for the power analysis was assumed as 6. The power (1 − β) was >0.9 at the level of α = 0.05 for both size of chalazion and SPEED score, and the sample size was sufficient.

Statistical analysis was performed with JMP Pro version 16 software (SAS, Cary, NC, USA). Data are shown as means ± SDs. All statistical tests were two-sided, and a *p* value of <0.05 was considered statistically significant.

## 3. Results

Patient demographics are shown in Table 1. Twenty-nine chalazia of 24 eyelids of 12 patients (14 chalazia of 14 eyelids of 6 patients in the MGX alone group and 15 chalazia of 14 eyelids of 6 patients in the IPL-MGX group) were enrolled in the study. No significant differences in parameters were detected between the two groups before treatment (Table 1).

### 3.1. Efficacy of IPL-MGX

The characteristics of the eyes in the IPL-MGX group and the control group before as well as 4 weeks after the final treatment are shown in Table 2. The size of chalazia and VAS score were significantly decreased (Figure 2, Table 2). Significant decreases in irregularity (*p* = 0.031), plugging, vascularity, CFS, meibum grade, the number of Demodex, diameter of chalazion, and VAS score (*p* < 0.001, respectively) were apparent at 4 weeks after the final treatment in the IPL-MGX group (Table 2). Meiboscore and Schirmer test value at 4 weeks after the final treatment did not differ significantly in the IPL-MGX group (*p* = 1.00, 0.76, respectively) (Table 2). The SPEED score was significantly reduced at 4 weeks after the final treatment in the IPL-MGX group (*p* = 0.031) (Table 3).

All of the parameters except meibum grading in the MGX alone group remained unchanged (Table 2). Comparing the IPL-MGX and MGX alone groups, the size of chalazia, VAS, plugging, vascularity, CFS, meibum grade, and number of Demodex were significantly improved (*p* < 0.001, respectively) (Table 2). 

### 3.2. The Number of IPLs Required to Improve the Chalazion

Among the background factors, only the size of the chalazion correlated with the number of IPL-MGX sessions for treatment. (Table 4). Among the parameters related to the meibomian gland and tear film, plugging, CFS score, meibum grade, the number of *Demodex*, and Schirmer value were positively correlated with the number of IPL-MGX sessions for the treatment of chalazia (Table 5).

### 3.3. Safety of IPL-MGX

There were no significant differences in visual acuity and intraocular pressure before and 4 weeks after the final treatment in either treatment group (Table 6). Lens opacity and fundus condition showed no change between before and 4 weeks after the final treatment in either treatment group.

## 4. Discussion

The safety and effectiveness of IPL combined with MGX: Although chalazion is a non-infectious granuloma that often resolves spontaneously without special treatment, it can recur frequently, resulting in refractory chalazion. In this study, we treated multiple recurrent chalazia in the IPL-MGX and the MGX alone groups without surgery or incision, and found that the IPL-MGX treatment significantly reduced the size of the chalazion and increased patient satisfaction. This is the first study to examine the usefulness of IPL-MGX without surgical treatment and to compare its efficacy with MGX alone. The results showed that the IPL-MGX group significantly improved the size of chalazion, subjective symptoms, MGD-related parameters, and number of Demodex compared to the control group. The treatment was safe and effective with no side effects.

The mechanism by which IPL was effective for chalazion may be due to the anti-inflammatory effect of IPL and the mechanism of chalazion by temperature rise [29], since chalazion is an inflammatory granuloma. In addition, the IPL-MGX group was able to suppress recurrence for 6 months, suggesting that the environment of the ocular surface, including the eyelid, improved and the recurrence of the chalazion itself was suppressed by the treatment with IPL not only in the meibomian glands affected by the chalazion but also in the entire eyelid. The larger the size of chalazion at baseline was, and the poorer the function and morphology of the meibomian gland was, the higher the number of IPL cycles which was required to cure the chalazion in our study. An average of 2.8 (maximum 4) IPL cycles were required for the treatment of recurrent chalazion, which is similar to the number of IPL cycles required for the treatment of MGD. 

### 4.1. Risk Factors for Chalazion

Blepharitis is the most common risk factor for chalazion. The probability of having a chalazion in the presence of blepharitis is 4.7 times greater than in the absence of blepharitis [3]. Posterior blepharitis describes inflammatory condition of the posterior lid margin, of which MGD is one possible cause [43]. Moreover, the previous report showed that MGD, dry eye, and blepharitis were the risk factors for recurrent chalazion [6,44]. Recently, the relationship between Demodex and chalazion has become a controversial topic. The rate of Demodex positivity is very high, ranging from approximately 70% to 90% in eyes affected by chalazion [4,45]. Demodex positivity is also associated with a high recurrence rate of chalazion [45]. Therefore, we investigated the meibomian-gland-related parameters, tear-film-related parameters and the presence of Demodex to determine the effect of IPL on multiple recurrent chalazia.

### 4.2. Compared to the Previous Results

In this study, three cases (50%) cured within 1 month; three (50%) cured after 1.5 months. In addition, all six cases had no recurrence 6 months after the end of the IPL treatment. A prospective randomized clinical trial reported that complete resolution rates in the surgical treatment and the triamcinolone acetonide injection groups were 79% and 81% [46]. The average time to resolution in the surgical treatment and the triamcinolone acetonide injection groups was 4 days and 5 days [46]. A single-center prospective randomized clinical study reported that the resolution rates in the surgical treatment, one triamcinolone acetonide injection, and hot compress groups were 87%, 84%, and 46% at the 3-week follow-up [47]. Patients with more than one chalazion on the same eyelid were excluded from the study. A prospective randomized multicenter treatment study reported that the resolution rates of single or multiple chalazia in the hot compresses alone, the hot compresses plus tobramycin, and hot compresses plus tobramycin/dexamethasone groups were 21%, 16%, and 18%, respectively, for 4–6 weeks of treatment [10].

### 4.3. Compared to the Conventional Therapies

Conventional treatments for chalazion include surgery, incision, warm compresses, lid hygiene, and topical steroid injection [1]. Non-ablative treatment takes longer (about 6 months on average) than surgery and incision. According to the previous literature, a meta-analysis on the recurrence rate of primary chalazion showed that the recurrence rate for curettage for chalazia was 0–16.7% and 0–27.3% for intralesional steroid injections [48]. Demodex blepharitis, MGD, and dry eye are also reported to be risks for multiple recurrent chalazia [34,44], and repeated surgery may further damage the meibomian glands, leading to decreased visual function [13] and eventual dry eye. In particular, the younger generation can be damaged cosmetically, mentally, and visually by multiple recurrences. Therefore, it is preferable to treat multiple recurrent chalazia without surgery and to treat subclinical/clinical MGD to prevent recurrence. We believe that proactive treatment for MGD, in addition to the local treatment of chalazion, can prevent future recurrence of chalazion and contribute to the patient’s quality of life and quality of vision.

### 4.4. Limitations

There are several limitations in this study. First, this was a retrospective study. Second, the sample size was small, although the number of cases was sufficient for the power analysis. Third, the number of IPL sessions was not identical from case to case due to the search for the optimal IPL protocol for chalazion. Fourth, no automatic procedure was used to determine effectiveness. Finally, the decision point is relatively subjective, as patient satisfaction with improvement is clinically important for chalazion. Further addition of cases and multicenter studies are desirable with a randomized controlled prospective study. Further studies for the investigation of IPL protocols for recurrent multiple chalazia based on MGD treatment are needed.

## 5. Conclusions

In conclusion, our study demonstrated that an average of 2.5 ± 0.8 chalazia per patient, 9.0 ± 5.2 mm in size, were improved with an average of 2.8 ± 1.3 IPLs, with no recurrence for 6 months. IPL was safe and effective as non-surgical treatment of multiple recurrent chalazia. IPL is probably the best treatment option for multiple recurrent chalazia in terms of prevention of recurrence in order to minimize loss of the meibomian glands.

## Figures and Tables

**Figure 1 jcm-11-05338-f001:**
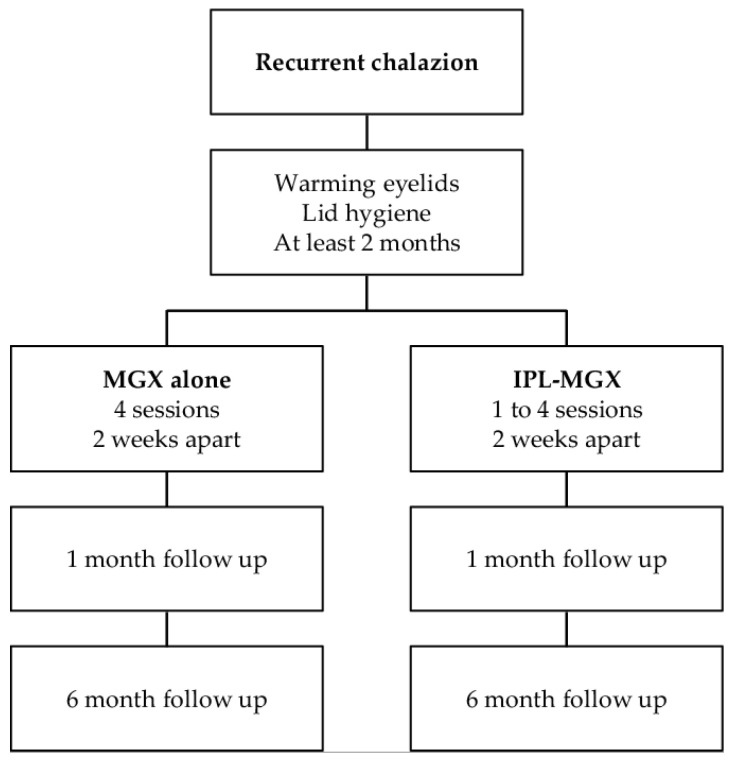
The Clinical Flow of this study.

**Figure 2 jcm-11-05338-f002:**
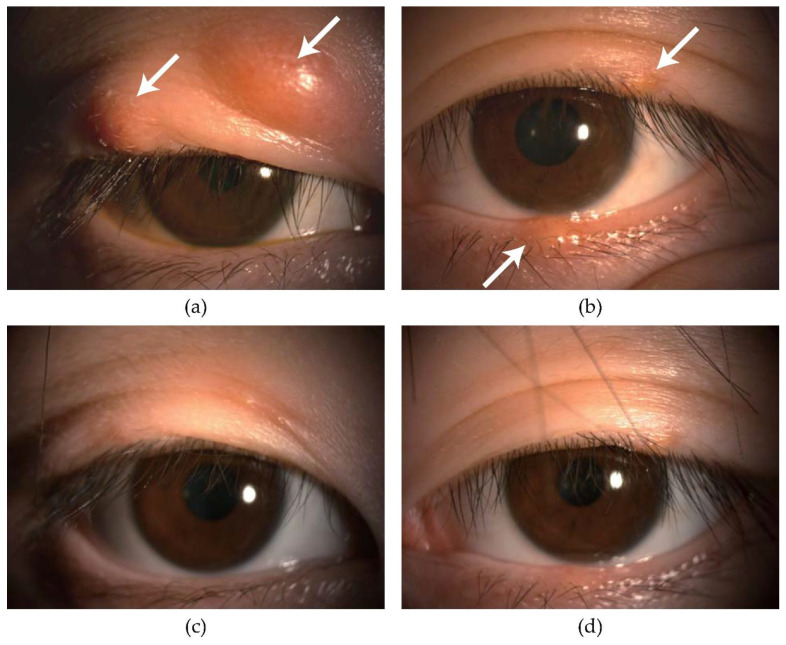
A 19-year-old female. Changes in chalazion in upper and lower eyelids before and after IPL. (**a**) Upper right eye. 18mm of chalazia (right white arrow) and 7 mm of chalazia (left white arrow) recurrenced. (**b**) Upper left eye. 4 mm of chalazia (upper white arrow) and 6 mm of chalazia (lower white arrow) recurrenced. (**c**) Two upper chalazia much improved. (**d**) Tow upper and lower chalazia much improved.

**Table 1 jcm-11-05338-t001:** Demographics of the patients with chalazion in Intense pulsed light (IPL)–meibomian gland expression (MGX) (*n* = 6) and MGX alone (*n* = 6) groups.

	IPL-MGX		MGX Alone		*p* Value
	Mean ± SD	(Range)	Mean ± SD	(Range)	
Age (years)	36.8 ± 12.1	(19–51)	37.7 ± 13.1	(22–54)	1.00
Number of chalazia	2.5 ± 0.8	(2–4)	2.3 ± 0.5	(2–3)	0.92
Number of eyelids with chalazia	2.3 ± 0.5	(2–3)	2.3 ± 0.5	(2–3)	1.00
Duration of pre-lid-warming (months)	30.9 ± 41.2	(0.5–104)	2.0 ± 2.1	(0.5–6)	0.29
Size of the largest chalazion (mm)	12.2 ± 5.6	(5–18)	10.5 ± 2.0	(7–12)	0.81

*p* values were obtained using Wilcoxon rank sum test.

**Table 2 jcm-11-05338-t002:** Characteristics of the patients with chalazion in intense pulsed light (IPL)–meibomian gland expression (MGX) (treatment group) (*n* = 12) and MGX alone (control group) (*n* = 12) groups before and four weeks after the final treatment session.

Characteristics		Baseline		*p* Value for IPL-MGX vs. MGX Alone	Post-Treatment		*p* Value vs. Baseline	*p* Value for IPL-MGX vs. MGX Alone
		Mean ± SD	(Range)		Mean ± SD	(Range)		
**Number of IPL for improvement**	**MGX alone**							
	**IPL-MGX**				2.8 ± 1.3	(1–4)		
**Plugging (0–3)**	**MGX alone**	2.7 ± 0.5	(2–3)	0.92	2.1 ± 0.9	(1–3)	0.063	<0.001 **
	**IPL-MGX**	2.6 ± 0.7	(1–3)		0.2 ± 0.4	(0–1)	<0.001 **	
**Vascularity (0–3)**	**MGX alone**	2.3 ± 0.8	(1–3)	0.77	2.3 ± 0.8	(1–3)	1.00	<0.001 **
	**IPL-MGX**	2.4 ± 0.8	(1–3)		0 ± 0	(0–0)	<0.001 **	
**Irregularity (0–2)**	**MGX alone**	0.9 ± 0.9	(0–2)	0.83	0.9 ± 0.9	(0–2)	1.00	0.27
	**IPL-MGX**	1.0 ± 0.9	(0–2)		0.5 ± 0.5	(0–1)	0.031 *	
**CFS (0–9)**	**MGX alone**	2.0 ± 0.7	(1–3)	0.54	1.7 ± 0.7	(1–3)	0.125	<0.001 **
	**IPL-MGX**	2.1 ± 1.5	(1–5)		0.2 ± 0.4	(0–1)	<0.001 **	
**Meibum grade (0–3)**	**MGX alone**	2.5 ± 0.5	(2–3)	0.57	1.9 ± 0.8	(1–3)	0.016 *	<0.001 **
	**IPL-MGX**	2.6 ± 0.7	(1–3)		0.3 ± 0.5	(0–1)	<0.001 **	
**Meiboscore (0–6)**	**MGX alone**	3.2 ± 0.9	(2–4)	0.88	3.2 ± 0.9	(2–4)	1.00	0.88
	**IPL-MGX**	3.4 ± 1.5	(2–6)		3.4 ± 1.5	(2–6)	1.00	
**Number of *Demodex***	**MGX alone**	3.1 ± 0.9	(2–4)	0.52	3.3 ± 0.8	(2–4)	0.50	<0.001 **
	**IPL-MGX**	2.8 ± 0.9	(2–4)		0 ± 0	(0–0)	<0.001 **	
**Schirmer test value (mm)**	**MGX alone**	11.7 ± 4.7	(5–20)	0.50	11.0 ± 4.2	(6–20)	0.30	0.75
	**IPL-MGX**	11.5 ± 7.5	(4–26)		11.5 ± 5.5	(6–20)	0.76	
**Size (diameter) of chalazion (mm)**	**MGX alone**	8.8 ± 2.4	(6–12)	0.47	9.2 ± 2.6	(6–12)	0.38	<0.001 **
	**IPL-MGX**	9.0 ± 5.2	(3–18)		0 ± 0	(0–0)	<0.001 **	
**VAS score (0–100)**	**MGX alone**	61.2 ± 22	(23–90)	1.00	67.3 ± 21.2	(30–90)	0.063	<0.001 **
	**IPL-MGX**	61.7 ± 23.8	(23–90)		0 ± 0	(0–0)	<0.001 **	

CFS, corneal and conjunctival fluorescein staining; VAS score, visual analogue scale score of ocular discomfort and foreign body sensation. *p* values were determined with Wilcoxon signed-rank test or Wilcoxon rank sum test (* *p* < 0.05, ** *p* < 0.001).

**Table 3 jcm-11-05338-t003:** Standard Patient Evaluation of Eye Dryness (SPEED) validated questionnaire score (0–28) of the patients with chalazion in intense pulsed light (IPL)–meibomian gland expression (MGX) (*n* = 6) and MGX alone (*n* = 6) groups before and four weeks after the final treatment session.

	Baseline		*p* Value for IPL-MGX vs. MGX Alone	Post-Treatment		*p* Value vs. Baseline	*p* Value for IPL-MGX vs. MGX Alone
	Mean ± SD	(Range)		Mean ± SD	(Range)		
**MGX alone**	11.8 ± 2.0	(9–15)	0.94	12.3 ± 2.2	(9–15)	1.00	0.003 *
**IPL-MGX**	11.2 ± 4.3	(4–15)		0 ± 0	(0–0)	0.031 *	

*p* values were determined with Wilcoxon signed-rank test or Wilcoxon rank sum test (* *p* < 0.05).

**Table 4 jcm-11-05338-t004:** Spearman’s correlation coefficient (ρ) and *p* values for the relation between baseline characteristics and the number of treatment sessions in the intense pulsed light (IPL)–meibomian gland expression (MGX) group (*n* = 6).

Characteristics	ρ	*p* Value
Age	0.12	0.82
Number of chalazia	0.66	0.16
Number of eyelids with chalazia	0.67	0.15
Duration of pre-lid-warming	0.62	0.19
Size of the largest chalazion	0.94	0.005 *
SPEED score at baseline	−0.03	0.95

* *p* < 0.05. SPEED score, Standard Patient Evaluation of Eye Dryness validated questionnaire score.

**Table 5 jcm-11-05338-t005:** Spearman’s correlation coefficient (ρ) and *p* values for the relation between baseline parameters and the number for treatment in the intense pulsed light (IPL)–meibomian gland expression (MGX) group (*n* = 12 eyes).

Baseline Parameters	ρ	*p* Value
Plugging	0.74	0.006 *
Vascularity	0.56	0.059
Irregularity	0.52	0.086
CFS	0.89	<0.001 **
Meibum grade	0.74	0.006 *
Meiboscore	0.57	0.051
Number of Demodex	0.73	0.007 *
Schirmer test value	0.61	0.034 *
VAS score	−0.24	0.45

* *p* < 0.05, ** *p* < 0.001. CFS, corneal and conjunctival fluorescein staining; VAS score, visual analogue scale score of ocular discomfort and foreign body sensation.

**Table 6 jcm-11-05338-t006:** Visual acuity and intraocular pressure of the patients with chalazion in intense pulsed light (IPL)–meibomian gland expression (MGX) (*n* = 12) and MGX alone (*n* = 12) groups before and four weeks after the final treatment session.

Characteristics		Baseline		*p* Value for IPL-MGX vs. MGX Alone	Post-Treatment		*p* Value for IPL-MGX vs. MGX Alone	*p* Value vs. Baseline
		Mean ± SD	(Range)		Mean ± SD	(Range)		
**LogMAR visual acuity**	**MGX alone**	−0.06 ± 0.07	(−0.18–0.00)	0.37	−0.07 ± 0.06	(−0.18–0.00)	0.38	0.50
	**IPL-MGX**	−0.07 ± 0.03	(−0.08–0.00)		−0.08 ± 0	(−0.08–0.08)		0.50
**IOP (mmHg)**	**MGX alone**	16.7 ± 1.4	(15–19)	0.93	16.6 ± 1.1	(14–19)	0.98	0.77
	**IPL-MGX**	16.8 ± 1.5	(15–19)		16.6 ± 1.7	(15–18)		1.00

LogMAR, logarithm of minimum angle of resolution; IOP, intraocular pressure. *p* values were determined with Wilcoxon signed-rank test or Wilcoxon rank sum test.

## Data Availability

The datasets generated during and analyzed in the current study are available from the corresponding author on request.

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
