# Peer review of "Therapeutic Efficacy and Safety of Intense Pulsed Light for Refractive Multiple Recurrent Chalazia"

_jcm, 2022, doi:10.3390/jcm11185338_

Round 1
Reviewer 1 Report
In this paper Arita & Fukuoka demonstrate that a combination of intense pulsed light and Meibomian gland expression can improve reduce chalazia severity as indicated by several metrics. I find the manuscript clear, the statistical tests appropriate, and the conclusions reasonable. But this article could be improved if the following minor comments are addressed.
Minor comments
Some grammatical errors in the text remain. For example, in the abstract (line 21): “There were no adverse effects occurred in both either groups.”
Similarly, some sentences are not clear: e.g. in line 22 “IPL-MGX was safe and effective for multiple and recurrent chalazia without surgery or curettage by improving meibomian gland function.” it is unclear what “by improving Meibomian gland function” is referring to.
The content at lines 135–139 belongs in results, not methods.
Author Response
Reviewer 1
In this paper Arita & Fukuoka demonstrate that a combination of intense pulsed light and Meibomian gland expression can improve reduce chalazia severity as indicated by several metrics. I find the manuscript clear, the statistical tests appropriate, and the conclusions reasonable. But this article could be improved if the following minor comments are addressed.
Response: We wish to express our appreciation for your helpful comments. Your concerns have been addressed accordingly.
Minor comments
Some grammatical errors in the text remain. For example, in the abstract (line 21): “There were no adverse effects occurred in both either groups.”
Response: Thank you for pointing this out. It has been corrected. (Line 21)
Similarly, some sentences are not clear: e.g. in line 22 “IPL-MGX was safe and effective for multiple and recurrent chalazia without surgery or curettage by improving meibomian gland function.” it is unclear what “by improving Meibomian gland function” is referring to.
Response: We have clarified the sentence in lines 22-23.
The content at lines 135–139 belongs in results, not methods.
Response: Thank you for your comment. Lines 143-152 describe the Power analysis method. Power analysis is usually described in the Methods section of the retrospective study.

Reviewer 2 Report
The authors evaluate the efficacy and safety of IPL with MGX in chalazia and they found that IPL + MGX improve MGD.
The paper is interest and described a novel topic. The authors should solve the comment propose in order to continue with the publication process.
2- In the title should be a mention added to MGX
13 - 29 chalazia of 12 patient? There was recurrence?
15 – one to four? There was disagreement per patient, why?
18 – include number data on abstract
21 – include extra info to the conclusion, not only effective and safe.
29 – poor scientific redaction
30 – add short info about physiopathology of risk factor in chalazion
35 – include additional reference for all this section
50 – add IPL side effects
60 – seems that your paper establishes a protocol, and it not the aim, rewrite.
70 – why a mixture of IPL and MGX versus MGX alone?
76 – inclusion criteria were no answer to all the medications?
83 – one to four sessions are heterogeneous, describe the reason and include in limitation
89 – very subjective decision point
93 – Flowchart seems that all patients follow the same protocol, clarify
96 – any automatic procedure could be possible?
147 – how does randomization performed?
153 – p value of 1 does not exist
175 – described first treatment group and second the control one
202 – described in detail safety section
206 – divide the discussion in topic sections
228 – include comparison with your results
241 – this paragraphs seems to be form and introduction
249 – this resume info could be described at the beginning of the discussion
262 – there are more limitation, subjective measurements, heterogeneity of the sample and the methodology …
267 – extend the conclusion with specific statements about the finding achieved
271 – ethics disclosure
276 – data disclosure
279 – update references prior to 2010 when possible
279 – use only indexed journals
Author Response
Reviewer 2
2- In the title should be a mention added to MGX
Response: Thank you for your comment. In this controlled study, we compared two groups: the IPL plus MGX group and the MGX alone group, and we supposed that IPL was effective because the IPL plus MGX group significantly improved chalazion in all parameters except irregularity of the lid margin, the meiboscore, and Schirmer value compared to the MGX alone group. Therefore, we did not put MGX in the title.
13 - 29 chalazia of 12 patient? There was recurrence?
Response: Thank you for your comment. All patients had multiple and recurrent chalazion as we described in the materials and methods section.
15 – one to four? There was disagreement per patient, why?
Response: Because this was an exploratory retrospective study. The number of IPL was different from each patient because we would like to end with the minimum number of IPL once the patient's chalazion improved. We added this point in the discussion section as one of limitations. (lines 292-293)
18 – include number data on abstract
Response: Thank you very much for your comment. Unfortunately, due to word limits in the abstract (200 words), we were unable to include them in the abstract; they are listed in Table 2 and Table 3.
21 – include extra info to the conclusion, not only effective and safe
Response: We have clarified the sentence in lines 22-23.
29 – poor scientific redaction
Response: Thank you for your helpful comment. We added the references below.
- F. Duarte, E. Moreira, A. Nogueira, P. Santos, and F. Azevedo, "Chalazion surgery: advantages of a subconjunctival approach," (in eng), J Cosmet Laser Ther, vol. 11, no. 3, pp. 154-6, Sep 2009, doi: 10.1080/14764170902902822.
- Y. Wong, G. S. Yau, J. W. Lee, and C. Y. Yuen, "Intralesional triamcinolone acetonide injection for the treatment of primary chalazions," (in eng), Int Ophthalmol, vol. 34, no. 5, pp. 1049-53, Oct 2014, doi: 10.1007/s10792-014-9904-1.
- W. Lee, G. S. Yau, M. Y. Wong, and C. Y. Yuen, "A comparison of intralesional triamcinolone acetonide injection for primary chalazion in children and adults," (in eng), ScientificWorldJournal, vol. 2014, p. 413729, 2014, doi: 10.1155/2014/413729.
- Goawalla and V. Lee, "A prospective randomized treatment study comparing three treatment options for chalazia: triamcinolone acetonide injections, incision and curettage and treatment with hot compresses," (in eng), Clin Exp Ophthalmol, vol. 35, no. 8, pp. 706-12, Nov 2007, doi: 10.1111/j.1442-9071.2007.01617.x.
- J. Ben Simon, N. Rosen, M. Rosner, and A. Spierer, "Intralesional triamcinolone acetonide injection versus incision and curettage for primary chalazia: a prospective, randomized study," (in eng), Am J Ophthalmol, vol. 151, no. 4, pp. 714-718.e1, Apr 2011, doi: 10.1016/j.ajo.2010.10.026.
- Y. Wu, K. A. Gervasio, K. N. Gergoudis, C. Wei, J. H. Oestreicher, and J. T. Harvey, "Conservative therapy for chalazia: is it really effective?," (in eng), Acta Ophthalmol, Jan 16 2018, doi: 10.1111/aos.13675.
30 – add short info about physiopathology of risk factor in chalazion
RESPONSE: We added the sentences in lines 29-32.
35 – include additional reference for all this section
RESPONSE: We added the additional references in lines 35-42.
50 – add IPL side effects
RESPONSE: We added the side effects by IPL in lines 57-59.
60 – seems that your paper establishes a protocol, and it not the aim, rewrite.
RESPONSE: We added the sentence in lines 69-70.
70 – why a mixture of IPL and MGX versus MGX alone?
RESPONSE: Because multiple and recurrent chalazia were considered a focal type of MGD, and IPL plus MGX treatment for MGD and MGX as a control group were used.
76 – inclusion criteria were no answer to all the medications?
RESPONSE: You are right. This is for people who have not responded to all previous treatments.
83 – one to four sessions are heterogeneous, describe the reason and include in limitation
RESPONSE: Because there had been no previous reports for this, it was unclear how many times it would be optimal. We added this in the limitation in lines 292-293.
89 – very subjective decision point
RESPONSE: We changed the sentence to clarify the endpoint more objectively in line 99-100.
93 – Flowchart seems that all patients follow the same protocol, clarify
RESPONSE: We clarified the flowchart in Figure 1.
96 – any automatic procedure could be possible?
RESPONSE: Thank you very much for your helpful comment. However, at this stage, it was difficult to use the automatic procedure. We added this point in the limitation in lines 293-294.
147 – how does randomization performed?
RESPONSE: This was a retrospective study. Therefore, this was not randomized. We deleted the word “randomly” in line 81.
153 – p value of 1 does not exist
RESPONSE: We confirmed the number of P value. It's possible to get an exact p-value of 1 on Wilcoxon rank sum test. We rewrote 1 to 1.00 in Table 1.
175 – described first treatment group and second the control one
RESPONSE: We added the treatment group and control group in Table 2.
202 – described in detail safety section
RESPONSE: We added Table 6.
206 – divide the discussion in topic sections
RESPONSE: We divided the discussion in topic sections as you suggested.
228 – include comparison with your results
RESPONSE: We included comparison with our results in lines 259-272.
241 – this paragraphs seems to be form and introduction
RESPONSE: We deleted this paragraph and added the sentence in line 52-54.
249 – this resume info could be described at the beginning of the discussion
RESPONSE: We moved this paragraph at the beginning of the discussion in lines 234-244.
262 – there are more limitation, subjective measurements, heterogeneity of the sample and the methodology …
RESPONSE: We added three more limitations in the discussion in lines 292-295.
267 – extend the conclusion with specific statements about the finding achieved
RESPONSE: We added the specific statements about the finding achieved in lines 301-303.
271 – ethics disclosure
RESPONSE: We added the ethics disclosure in line 313-315.
276 – data disclosure
RESPONSE: We added the data disclosure in line 318-319.
279 – update references prior to 2010 when possible
RESPONSE: Since chalazion is a classic disease, many important papers on conventional treatment were written before 2010. We could not remove these references.
279 – use only indexed journals
RESPONSE: Thank you for your comment. We reviewed all of the references.

Round 2
Reviewer 2 Report
Comments solved